# The Role of Bacteriophages in the Gut Microbiota: Implications for Human Health

**DOI:** 10.3390/pharmaceutics15102416

**Published:** 2023-10-03

**Authors:** Stephen C. Emencheta, Chinasa V. Olovo, Osita C. Eze, Chisom F. Kalu, Dinebari P. Berebon, Ebele B. Onuigbo, Marta M. D. C. Vila, Victor M. Balcão, Anthony A. Attama

**Affiliations:** 1Department of Pharmaceutical Microbiology and Biotechnology, University of Nigeria, Nsukka 410001, Nigeria; stephen.emencheta@unn.edu.ng (S.C.E.); osy.eze@unn.edu.ng (O.C.E.); okoreflora@gmail.com (C.F.K.); ebele.onuigbo@unn.edu.ng (E.B.O.); 2VBlab—Laboratory of Bacterial Viruses, University of Sorocaba, Sorocaba 18023-000, Brazil; marta.vila@prof.uniso.br (M.M.D.C.V.); victor.balcao@prof.uniso.br (V.M.B.); 3Department of Microbiology, University of Nigeria, Nsukka 410001, Nigeria; chinasa.olovo@unn.edu.ng; 4Department of Biology and CESAM, University of Aveiro, Campus Universitário de Santiago, P-3810-193 Aveiro, Portugal; 5Department of Pharmaceutics, University of Nigeria, Nsukka 410001, Nigeria; 6Institute for Drug-Herbal Medicine-Excipient Research and Development, University of Nigeria, Nsukka 410001, Nigeria

**Keywords:** bacteriophages, gut microbiota, human health, therapeutic applications

## Abstract

Bacteriophages (phages) are nano-sized viruses characterized by their inherent ability to live off bacteria. They utilize diverse mechanisms to absorb and gain entry into the bacterial cell wall via the release of viral genetic material, which uses the replication mechanisms of the host bacteria to produce and release daughter progeny virions that attack the surrounding host cells. They possess specific characteristics, including specificity for particular or closely related bacterial species. They have many applications, including as potential alternatives to antibiotics against multi-resistant bacterial pathogens and as control agents in bacteria-contaminated environments. They are ubiquitously abundant in nature and have diverse biota, including in the gut. Gut microbiota describes the community and interactions of microorganisms within the intestine. As with bacteria, parasitic bacteriophages constantly interact with the host bacterial cells within the gut system and have obvious implications for human health. However, it is imperative to understand these interactions as they open up possible applicable techniques to control gut-implicated bacterial diseases. Thus, this review aims to explore the interactions of bacteriophages with bacterial communities in the gut and their current and potential impacts on human health.

## 1. Introduction

The ubiquitous nature of microorganisms extends to their presence in the gastrointestinal tract (GIT) [1,2], where they play a host of unique physiological characteristics and functions which contribute to the overall well-being and health [2], including, but not limited to, their help in digestive and metabolic processes [3], gut barrier protection [4], essential vitamin production [5], and the immune system [6].

The gut microbiota is influenced by a potential compounding combination of different factors determining the microbial makeup [7]. These include diet or nutrient type (e.g., plant and fiber-based diets promote the selection/presence of beneficial microorganisms, unlike chemically processed and sugary diets, and breast milk has significant influence on the infant microbiota in the first few years) [7], external chemicals (e.g., antibiotics) [8,9], genetic/hereditary profile (although largely unclear) [10], age [9], environmental pollutants/chemicals (e.g., agrochemicals), medical conditions (e.g., celiac and IBDs), and lifestyle [9].

Pathogenic/unhealthy gut microorganisms can hardly ever be eradicated and can even be potentially beneficial; however, maintenance of the equilibrium between the healthy and harmful gut microorganisms is essential and, through several mechanisms, is related to well-being and overall health and the prevention, development, and management of health/medical conditions [11]. Specifically, IBDs often result from the immune response to the dysbiosis of the gut microbiota, increasing the potentially pathogenic microorganisms (e.g., *Pseudomonadota* and *Enterobacteriaceae*) and decreasing the beneficial microorganisms (e.g., *Lactobacillaceae* and Bifidobacteria) [11]. Accumulative evidence has shown that disruption of the metabolites and the gut microbiota can influence insulin sensitivity [12], optimal brain and central nervous system (CNS) (which, of course, controls most of the GIT physiology) functions, and emotional behaviors [13,14]. High blood pressure, a top risk factor for heart disorders, can be initiated by disruption of the gut microbiota [15]. Also, specific microbial genes are associated with the generation and accumulation of Trimethylamine N-oxide (TMAO), a risk factor for cardiovascular diseases, especially at high concentrations [16,17]. The gut microbiota can be described as the “other organ”, contributing significantly to nutrient and energy acquisition and regulation. Thus, dysbiosis can negatively affect the above functions, triggering excessive tissue deposition and obesity [18].

One prominent and easily the most abundant life forms are the bacteriophages [8]. Bacteriophages have found applications in diverse fields and have peculiar characteristics in clinical practice against pathogens, including their respective specificity against limited bacterial genera and species [19,20]. They are also minimally allergenic with “little or no” side effects [21]. They form a major part of the gut microbiota and constantly interact with host bacterial strains [4]. Specifically, they naturally aid in the regulation/modulation of the gut bacterial population, acting as natural predators, selectively infecting and killing specific bacterial strains and, hence, aiding in maintaining the gut microbiota and its overall composition [22]. Also, specific interactions, which release cytokines and other immune molecules, contribute to the modulation of the immune system [4]. Furthermore, they significantly contribute to human metabolic processes and the general well-being in human health [4]. Currently, the diverse interactions of bacteriophages and their host bacteria and the implications for human health are poorly understood, and neither is the existing literature aggregated enough. While science struggles to keep up with the current and emerging interactions between gut viruses and the corresponding host bacteria strains [8], this review aims to discuss bacteriophages’ interactions with bacterial communities in the gut and their current and potential impacts on human health. In doing this, we shall discuss the basic concepts of bacteriophages, their characteristics, life cycle, and classifications, as well as the composition and functions of the gut microbiota. Next, we describe the impacts and roles of bacteriophages in maintaining the balance of the gut microbiota and their current and potential therapeutic applications. Finally, future research directions concerning the interactions between gut bacteriophages and the bacterial communities in the gut are discussed. The review method involved an overview of recently (2013–2023) published study reports using the relevant databases, including PubMed, Scopus, Google Scholar, and Thompson Reuters. The study search used keywords including “Bacteriophages”, “Gut Microbiota”, and “Human Health”, and the application of Boolean connectors were needed. Subscription-based articles were, however, not considered in the review.

## 2. Gut Microbiota and Composition

The human gastrointestinal tract is a tube-like organ originating from the mouth, transversing the esophagus, stomach, and small and large intestines and terminating in the anus. These organs harbor a large number and numerous classes of microorganisms that make up the gut microbiota (Figure 1), including bacteria [23], archaea, fungi, protozoa, and viruses (virome), including bacteriophages [24,25,26,27,28,29,30,31,32]. Increasing knowledge of the gut microbiota is enhanced by advancements in biotechnological tools and shows that the viral population in the gut could easily match or exceed those of the bacteria, which is against earlier beliefs [32]. They consist of diverse viral types, including eukaryotic (10%) and prokaryotic viruses (90%) [32] and the DNA and RNA viruses [33]; however, the small amount of literature on gut RNA viruses could be attributed to their instability in cultures [34] and the individual specificity of the gut viral composition, taking into account location, lifestyle, diet, age, and other factors [32,35].

## 3. Functions of the Gut Microbiota

Studies have established the link between the gut microbiota and human health. Some essential functions of the microbiota include their effects on the host’s nutrient metabolism, immunomodulation, bioprotection against invading pathogens, effects on xenobiotics and drug metabolism, and maintenance of mucosal integrity.

### 3.1. Effects of Gut Microbiota on Host Metabolism

The microbial communities in the gut are implicated in the breakdown of complex macromolecules, such as carbohydrates and other polymers, into readily absorbable forms, yielding short-chain fatty acids (SCFAs) as the major products [36], which not only act as a source of energy to cells but perform other essential functions. In addition to the generation of energy to the hosts, they equally contribute to the prevention of the accumulation of toxic substances [37]; lipid metabolism, in the synthesis, biotransformation, and biodegradation of dietary lipids [38]; protein metabolism and the breakdown of undigested proteins in the distal part of the gut to generate important metabolites using the aid of their specialized enzymes [39,40,41,42]; the synthesis of vitamins, such as vitamin K [43]; the biotransformation of bile acids [44]; and the conversion of inactive phenolic compounds into biologically active compounds [45].

### 3.2. Immunomodulatory Potential of Gut Microbiota

The human immune system and the gut microbiota are in constant interaction, aided by having a more significant percentage of the human immune system and the microbiota residing in the gut environment. The diverse nature of the microbiota community in the gut is considered the most significant driver for the development and maturity of innate and adaptive immunity [46,47,48]. As part of innate immunity, the gut environment favors obligate anaerobes, especially members of the Bacillota and Bifidobacteriaceae, while discouraging/suppressing the growth of facultative anaerobic pathogens [49]. Also, the innate immune system uses pattern recognition receptors (PRRs) [50] to identify both microbial antigens [51], triggering an immediate immunological response [49]. In adaptive immunity, the gut microbiota stimulates the B-cells to produce copious amounts of antibodies [52] and promotes the differentiation of T-cells into the necessary arsenals of cellular immunity [53]. Further to these interactions, the flexibility and specificity of adaptive immunity are responsible for the recognition and subsequent destruction of potential pathogens or other foreign bodies, while selectively escaping the beneficial/commensal microbes. The relationship between gut microbes and the immune system is the foundation of immune homeostasis [54].

### 3.3. Bioprotection against Invading Pathogens

The gut microbiota co-evolves with the host and establishes a mutualistic relationship, with the host providing a stable habitat for microbial sustenance and the microbes providing the host with beneficial physiological functions essential for the overall well-being of the host [55]. The host’s protection from pathogenic infection is crucial through colonization resistance [56,57], which often deprives a potential pathogen of the leverage to colonize and establish infection. This bioprotective function can be achieved by several means: competition and depletion of nutrients starving off the pathogen(s), generation of inhibitory short-chain fatty acids (SCFAs) and secondary bile acids, interference with the expression of colonization factors (quorum-sensing signals), and the production of antibacterial peptides, e.g., bacteriocins or microcins [58].

### 3.4. Effect on Xenobiotics and Drug Metabolism

Xenobiotics, which are foreign exogenous or endogenous substances, when accumulated to toxic levels in the body, can expose the body to deleterious health challenges [59,60]. While exogenous xenobiotics include drugs, food additives such as artificial sweeteners, and inhaled pollutants such as pesticides, insecticides, and cosmetics, endogenous xenobiotics include bile acids, certain fatty acids, eicosanoids, and steroids [61]. The effects of the gut microbiota on xenobiotics/drug metabolism can be beneficial or harmful to the body. Although the liver is primarily charged with dissipating the xenobiotic residues in the body, the gut microbiota also influences xenobiotics/drug metabolism. The human genome project, which birthed the human microbiome project (HMP) [62], and the omics, especially pharmacomicrobiomics, produced significant findings on the importance of the gut microbiota on xenobiotics/drug metabolism [63,64]. Specifically, most orally administered hydrophilic drugs are converted to easily absorbable hydrophobic metabolites by the gut microbiota, enabling systemic circulation [65]. The influence on xenobiotics/drug metabolism can be by direct action on the drug or via the production of enzymes capable of drug biotransformation of the medications into more active, less active, inactive, or more toxic metabolites [66,67]. It can also indirectly influence drug biotransformation through modification of the amount of the gut/liver-produced metabolizing enzymes [68], the release of competitive metabolites competing with drugs for the same metabolizing enzymes [69], and the activation of inactive drug metabolites already secreted [70].

### 3.5. Protection of Mucosal Integrity

Maintaining the gut mucosal integrity is the core responsibility of the intestinal epithelial cells (IECs), which invariably play a vital role in maintaining the host’s health. The epithelial cells can segregate between the gut microbiota and the pathogen through sensing and microbial recognition, achieved through continuous crosstalk between the IECs and the gut microbiota [71]. The IECs, upon sensing, can secrete mediators leading to inflammation induction or immune tolerance via the stimulation or suppression of immunocompetent cells, respectively [71]. Disrupting the effective structural and functional equilibrium between the IECs and the gut microbiota can impair the mucosal integrity with attendant adverse effects [72,73,74].

## 4. Bacteriophages: Classification, Life Cycle, and General Mechanism of Action

Bacteriophages are a group of abundant nano-sized viruses that propagate by predating the bacterial population. They are structurally simple and often made up of the head or capsid, tail, and fiber regions. Bacteriophage classification (Table 1) can be performed based on three (3) significant characteristics: morphology, genomic properties, and life cycle. Regarding morphology, Caudovirales are the most common, prominent, and studied family and possess long, non-enveloped tails [75]. Their subdivisions are Myoviridae, with contractile, complex baseplate long flexible tails; Podoviridae, with short noncontractile tails; and Siphoviridae, with noncontractile and rigid/non-flexible tails [75]. Filamentous bacteriophages have long, flexible, and filament-like structures with the genetic material inside a protein coat that is part of the structure [76]. Tectiviridaes possess a “tectiviral”-appearing lipid envelope surrounding an icosahedral head [77]. Inoviridaes have long, flexible, helical, and icosahedral heads (encapsulating the genetic material) [78]. Leviviridaes (RNA phages) are small-sized and possess a single-stranded RNA genome in an icosahedral capsid [79]. Microviridaes are non-enveloped and small-sized and have a single-stranded DNA genome in an icosahedral capsid [80]. Pleolipoviridaes are pleomorphic; thus, they can be filamentous, spherical, or irregular and possesses a lipid-containing envelope [81]. Regarding genomic properties, DNA phages can be double-stranded DNA (dsDNA)-based genome phages or single-stranded DNA (ssDNA)-based genome phages [82]. Also, RNA phages can be double-stranded RNA (dsRNA)-based genome phages or single-stranded RNA (ssRNA)-based genome phages [83,84]. A specific group of RNA Phages, retroviruses, can convert their RNA genome into DNA via reverse-transcriptase enzymes, including DNA polymerase and ribonuclease H (RNase H) [85]. Circular replicating phages possess circular genomes and replicate independently of the host cell’s replication machinery using a circle replication mechanism, termed rolling circle replication (RCR) [86]. Temperate (pseudo-lysogenic) phages can be lytic, causing host cell death, or lysogenic, becoming dormant within the host after integration into the host DNA [8]. Finally, virulent phages are strict lytic phages that replicate within the host cell, eventually leading to cell lysis and the release of daughter phage particles [8]. Temperate, lysogenic, and lytic processes classify phages according to their life cycle [8].

Lysogenic phages replicate their genomes alongside the host bacterial chromosomes, either integrated into the host’s chromosome or in a free, plasmid-like state, forming a long-term stable coexistence with the host until induced by appropriate environmental conditions [87,88]. This state is also referred to as the prophage state, and they remain dormant until the induction of the lytic cycle. In the lytic cycle, the phage starts the production of new viral progeny immediately after infection and releases them by lysing the host [89]. A virulent or lytic phage subverts the cellular apparatus of its bacterial host for multiplication, typically culminating in cell lysis and the release of progeny virions [88]. The lysis–lysogeny transcriptional switch controls phage entry into the lytic or the lysogenic cycle and uses different mechanisms. For instance, Erez et al. [90] reported phages of *Bacillus* spp. using a peptide-based arbitrium communication system in deciding whether to enter the lytic or lysogenic life cycle (Figure 2).

The rate-limiting step in phage bacteria destruction, however, depends on the rate of adsorption onto the bacterial surface [91]. Also, the fate of the host bacteria is affected by the release of phage proteins, including holins and endolysins, which assist in compromising the bacterial cell wall externally or the internal breaking of the cell wall to release phage particles [92,93,94].

## 5. Impact and Roles in Maintaining the Balance of the Gut Microbiota

The human gut is densely populated with microorganisms at different concentrations that constantly interact [95]. Although the ratio of virotypes to species-level bacterial phylotypes in the ocean is determined to be greater by a factor of 10 (i.e., 10:1), in the gut, a closer 1:1 ratio is observed [96,97]. Compared with the bacterial makeup of the gut microbiome, little is still known about the composition and physiological function of the phage components (the phageome) in human gut populations [32]. Before now, owing to the limitations in techniques and toolkits available for the study of the phageome, much was not known in this field as only direct observation with the counting of virus-like particle (VLP) methods was prevalent for their investigation. These were performed with epi-fluorescence and transmission electron microscopy [32]. The isolation of bacteriophages infecting specific host strains in culture was also employed. With these microscope-based methods, the immense diversity of viral morphotypes per individual was used to estimate the total counts of bacteriophages in the colonic mucosa, cecal components, and fecal samples of humans. These were estimated to be between 10^9^–10^10^ VLPs per gram, revealing the *Caudovirales* order, represented by *Myoviridae*, *Podoviridae*, and *Siphoviridae* as the most prevalent. However, these methods do not appropriately characterize the gut phageome, as many of the bacteria in the distal gut, such as Ruminococcaceae and Lachnospiraceae, are challenging to manage or culture in the laboratory [32,98,99]. Thus, the available collections of phage strains of human fecal samples do not reflect the true diversity of human gut bacteriophages. The complexity and abundance of human gut bacteriophage populations became more evident with emerging technologies such as high-throughput metagenomic sequencing technology [32,100]. The metagenomic analysis of fecal viromes carried out by Reyes et al. [97] showed that about 81–93% of bacterial viruses in the gut are novel and can be neither assigned a taxonomic position nor linked to a bacterial host [101]. Other studies on the gut phageomes of healthy humans also established that these viruses are specific to individuals and have minimal overlap [102]. Thus, substantial diversity of these phageomes exists between and among individuals; however, temporary stability is observed [97,103]. Minot and colleagues reported that most of the gut bacteriophages could persist for extended periods due to the lysogenic interactions with corresponding hosts, thus having a much slower evolution rate than the few lytic phages present [101,103]. Moreover, phages interact specifically with a single bacteria strain, and studies have also postulated that in the human gut, the phage–bacteria ratio is maintained at a ratio of 1:1 [101,104].

The imbalance in intestinal microecology could lead to several systemic diseases [105,106,107]. Phages are implicated in the balance or imbalance of the intestinal microecology and can affect human health either directly, by preying on the ecological landscape of the bacterial hosts, or indirectly, by influencing the immune system or metabolic pathway. Since the activities of phages alter the number and characteristics of the host bacteria, it is imperative to effectively and constantly regulate the relationship between these microbial species, as this not only keeps one in good health but can also reverse diseases [107].

## 6. Potential Therapeutic Applications

Many pathogenic and opportunistic bacteria are inherent in the gut system and have led to gut disease initiation, development, and advancement. Phage therapies are, however, developed and targeted at these bacteria. Potential applications of phages for preventive and curative purposes have produced single phages, cocktails, genetically modified phages, and even a combination with some other therapeutics, including antibiotics and probiotics [108]. The extensive literature review revealed that studies are developing therapies against the most prevalent bacteria associated with gut system infections, including *Vibrio* spp., *Escherichia coli*, *Clostridioides difficile*, *Salmonella* spp., *Fusobacterium nucleatum*, *Shigella* spp., *Klebsiella pneumoniae*, *Listeria monocytogenes*, *Ruminococcus gnavus*, and *Campylobacter* spp. Though most are still in the in vitro investigation, preclinical, and clinical trial stages, diverse potential applications have been found for many bacteriophage species, and they serve as potent therapeutic alternatives to managing multiple-drug-resistant strains [109]. These studies have enabled the identification and profiling of phages individually and in combinations using different dosage forms and delivery systems. The studies in the literature involving the therapeutic application of phages targeted against these implicated gut diseases are reported in the following section and summarized in Table 2.

### 6.1. Therapeutic Applications of Bacteriophages against Pathogenic Vibrio spp.

Considering the already established potential of phage therapy in gastrointestinal (GI) disorders, Jaiswal et al. [109] evaluated the in vitro and in vivo (using rabbits) therapeutic efficacy of a pure lytic vibriophage cocktail against the *V. cholerae* strain MAK 757 (ATCC 51352) that is implicated in cholera. The results of the lytic effects of individual vibriophages, B1, B2, B3, B4, and B5, applied singly and in combination as a cocktail showed synergistic effects, with the cocktail outperforming each of the individual phages. A comparison of the therapeutic management of orally induced *V. cholerae* infection in mice using a phage cocktail, conventional antibiotics (ciprofloxacin), and an oral rehydration system by Jaiswal et al. [110] showed that although the antibiotics had a significantly better anti-*V. cholerae* effect, the phage cocktail presented a significantly better safety and specificity profile and, thus, was more reliable in managing the infection. Jun et al. [111] showed a 74% (20 out of 27) lysis of multi-resistant *Vibrio parahaemolyticus*, a marine bacterium implicated in gastroenteritis and transmitted via raw oyster consumption, with optimal phage protein achieved following immediate phage therapy after infection. Yen et al. [112] utilized the potential prophylaxis of a cocktail of three virulent *V. cholerae*-specific phages, ICP1, ICP2, and ICP3, with specific effects against *V. cholerae* domiciled in the small intestine. From the in vivo results, the 24-h oral administration of the phages before *cholera* infection reduced intestinal tract colonization, thereby preventing cholera-like diarrhea. In addition to successfully demonstrating prophylaxis in *V. cholerae*-induced diarrhea, phage resistance was also not observed in the *V. cholerae* colonies.

### 6.2. Therapeutic Applications of Bacteriophages against Pathogenic Escherichia coli

Nasr-Eldin et al. [113] isolated and characterized highly stable *Siphoviridae* and *Podoviridae* phages for their lytic potential against *Escherichia coli*-causing gastrointestinal diseases using phage and *E. coli* incubation in a high saline environment. The characterizations, including the host range and synergistic profile, suggested that these bacteriophages were ideal candidates for therapeutic use. Similarly, Abdulamir et al. [114] targeted *E. coli* strains implicated in human gastroenteritis using a cocktail of 140 specific lytic phages administered to mice via their drinking water, oral injection, or vegetable capsules. The results revealed that the group receiving phage therapy via vegetable capsules obtained the least positive fecal cultures. While the peak reduction in *E. coli* was seen between 5–10 days post phage feeding, the second best-performing group following the phages’ administration was the group treated via the drinking water, with the study providing insight into the possible use of phage feed as a biocontrol for eliminating *E. coli* from animal intestines. Also, Abdelaziz et al. [115] isolated, characterized, and reported the broad-coverage lytic efficacy of phage phPE42 against *E. coli* clinical isolates implicated in gastrointestinal tract infections in an in vivo experiment. Bourdin et al. [116] acknowledged the downside of phage species and strain specificity and underlined the need for page therapies with broad host ranges. Also, the need to develop better-targeted phage therapies for disease conditions is needed. In treating childhood diarrhea-associated *E. coli* infection, they obtained and tested the lytic effects of 89 T4-like phages against four (4) batches of *E. coli* isolates. The result revealed that specific phage therapies for the tested pathogens were difficult and complex owing to the geographical, epidemiological, and time differences, thus recommending the need to identify flexible and specific therapeutic phages. With over 80% of traveler (TD) and childhood diarrhea cases caused by a variety of enteropathogens, and with multi-resistant *E. coli* usually responsible for about 30–40% [117], Aleshkin et al. [117] developed and assayed a phage cocktail for prophylaxis against TD caused by *E. coli*, *Shigella flexneri*, *Shigella sonnei*, *Salmonella enterica*, *Listeria monocytogenes*, and *Staphylococcus aureus*, and obtained specific prevention effects against TD caused by *E. coli*. In a similar experiment, Vahedi et al. [118] assayed the potential of combining a specific bacteriophage and antibiotics targeted against enteropathogenic *E. coli* (EPEC), using both in vitro and in vivo models. The in vitro study showed that 10^6^ PFU/ml of the phage eliminated EPEC from infected HEp-2 cells. In vivo, administration of the phage:antibiotic combination presented a total reduction in EPEC after 24 h and is attributed to the potentiation effect of both the antibacterial agent and the phage. However, slight weight loss was observed in the mice, possibly due to the adverse impact of antibiotics on the microbiota. However, Sarker et al. [119], in their work against acute bacterial diarrhea in children, used safe, orally administered species-specific T4-like coliphages in human subjects and showed no improvement in diarrhea symptoms, which was attributed to insufficient phage coverage and low *E. coli* titers. However, they assume that higher oral phage doses might be necessary to obtain the desired outcome, which triggers the need for more knowledge using in vivo phage–bacterium interaction strategy to understand *E. coli* propagation in childhood diarrhea. Galtier et al. [120] utilized three virulent bacteriophages in therapy against the colonization of adherent invasive *E. coli* (AIEC) strain LF82 implicated in inflammatory bowel diseases, Crohn’s disease, and ulcerative colitis and showed sufficient phage replication in the ileum, cecum, and colon following murine gut analysis. A single day of treatment with the bacteriophages administered to LF82-colonized AIEC strain CEABAC10 transgenic mice, which express the human carcinoembryonic antigen-related cell adhesion molecule 6CEACAM6 glycoprotein receptor for AIEC, revealed a notable decrease in the AIEC count. Over two weeks of continuous treatment resulted in the absence of colitis symptoms in mice colonized with the bacterial strain. Cieplak et al. [121] demonstrated the safety of phages in relation to their ability to induce dysbiosis following a comparison with antibiotics in the in vitro decolonization of *E. coli* populations in the small intestine. From the study, unlike the synthetic drug, the phage preparation had a targeted lytic effect on the *E. coli* populations. It impacted no other commensal bacteria used in the study, thus supporting its application in personal medicine, as characterized by its targeting of the bacteria of interest and evasion of dysbiosis induction. The overall efficacy of an ascertained safe commercial phage cocktail, PreforPro^®^, on the gut microbiota and markers of intestinal and systemic inflammation in a healthy human population was studied by Gindin et al. [122]. Twenty-eight (28) days of phage consumption did not alter the normal gut microbiota of most individuals but significantly reduced the target *E. coli* population and the pro-inflammatory cytokine interleukin 4 (Il-4) responsible for inflammatory reactions in the gastrointestinal tract. Shiga-toxin-producing *E. coli* is implicated in severe, difficult-to-treat infections. The safety and tolerability of PreforPro^®^ were reported by Grubb et al. [123]. They demonstrated the possible enhancement of the combined impacts of a probiotic microorganism and phage (*Bifidobacterium lactis* BL04 + PreforPro^®^) on gastrointestinal discomfort and stool consistency in a healthy adult population. They presented significant improvement in gastrointestinal symptoms over four weeks of therapy administration without disruption of the gut microbiota. There was also an associated increase in the relative abundance of some microorganisms, including *Lactobacillaceae*. Thus, the study suggests a strong connection between phage and the use of probiotics in improving the microbiota of the intestinal environment. The study by Alomari et al. [124] also further supported the simultaneous administration of bacteriophages and probiotics. They administered *Lactobacillus* spp. and phages of pathogenic *E. coli* combinations in suppositories in treating diarrheal calves. They reported a reduction in the diarrheal symptoms following therapy, complete elimination 24–48 h post-therapy, and a significant increase in the body weight of the treated calves compared with the control. Hsu et al. [95] proposed the use of genetics-based anti-virulence mechanisms in neutralizing the expression of the bacterial toxin and minimizing resistance as a better opinion to the conventional antibacterial approach. Unlike the conventional mechanism of bacterial lysis, temperate phages can be genetically engineered and integrated into the bacterial chromosome, and they are capable of neutralizing targeted gut bacterial toxins, impeding the virulence factors by modifying bacterial function at the genetic level; thus, they are good candidates for this therapy [125]. Hsu et al. [95] and Hsu et al. [125] utilized temperate phages capable of self-integration into the bacterial genome in in vivo and in situ studies, respectively, and both reported significant repression of the Shiga toxin secretion from *E. coli* in the mammalian gut. Cepko et al. [126], using a mouse model, isolated a strictly lytic phage that kills strains of enteroaggregative *E. coli* associated with both acute and chronic diarrhea [127]. A single dose of the phage one day post-infection-administration significantly reduced the bacterial count without altering microbiota diversity. Green et al. [128] explored a location-targeting mechanism aimed at enhancing the phage specificity and lytic cycle and treating gut infection, having observed that invasive pathobionts could reside deep within the mucosal epithelium of the gastrointestinal tract. With the ability to bind to heparan-sulfated glycans on the epithelial surface, the bacteriophage can position itself close to the target host. Here, phage HP3 showed lytic activities toward *E. coli* ST131 in vitro in a murine sepsis model. However, it proved ineffective in similar activities in the murine intestinal tract. The absence of lytic abilities in the latter was attributed to the intestinal mucins. Under a simulated intestinal environment, a podovirus phage isolated from wastewater, while not altering the intestinal microbiota compared with antibiotics, proved to be more effective due to its inherent ability to bind to heparan-sulfated proteoglycans on the surface of intestinal epithelial cells, thus strongly suggesting this feature to be responsible for the targeted lytic activity against the host bacterium.

### 6.3. Therapeutic Applications of Bacteriophages against Pathogenic Clostridioides Difficile

As an alternative to the use of antibiotics in treating GIT dysbiosis due to *C. difficile* infection (CDI), Nale et al. [129] assayed individual phages and a phage cocktail containing different phage combinations for their synergistic potential in the adequate clearance of *C. difficile*. Results obtained after 36 h post-infection supported the potential application of phage combinations for the targeted eradication of CDI and also concluded that specific phage combinations caused the complete lysis of *C. difficile* in vitro and prevented the appearance of resistant strains. Selle et al. [130] attempted to repurpose the endogenous type I-B CRISPR-Cas system in *C. difficile* as an antimicrobial agent through the use of bacteriophage capable of expressing a self-targeting CRISPR that redirected endogenous CRISPR-Cas3 activity against the bacterial chromosome and demonstrated that a recombinant bacteriophage expressing bacterial-genome-targeting CRISPR RNAs had significant lytic activities against *C. difficile* in both in vitro and mouse models. The study suggested that phage-delivered programmable CRISPR therapeutics have the potential to increase safety, specificity, and efficacy in complex gut microbial communities and offer a novel mechanism for the treatment of gut pathobionts.

### 6.4. Therapeutic Applications of Bacteriophages against Pathogenic Salmonella spp.

Using animal models, Dallal et al. [131] analyzed phage SE20 active against *Salmonella enteritidis*, a Gram-negative bacterium that occurs mainly in human gastroenteritis and is often implicated in salmonellosis, a disease commonly caused by the ingestion of animal-derived products, mainly poultry products (meat and eggs), that are significant carriers of *Salmonella* spp. [132]. The in vivo study revealed that a single dose (2 × 10^9^ PFU/mL) of the phage isolate provided a targeted and prophylactic effect against *S. enteritidis*. Additionally, while there was no bacterial resistance over twelve months of observation, compared with the animal groups receiving phage therapy, the test synthetic antibiotic used caused apparent weight loss in the administered experimental mice. Moye et al., 2019 [133] demonstrated that the direct ingestion of phages against *Salmonella* could enhance intrinsic gut resilience and provide protection against *Salmonella*-induced foodborne diseases. In their study, Simulator of the Human Intestinal Microbial Ecosystem (SHIME), a system aimed at exploring the potential of phage cocktails termed foodborne outbreak pills (FOPs) to eliminate foodborne pathogens and maintain the balance of the host microbiome, effectively depopulated the *Salmonella* without any distortion of stability of the gut microbiota. Thanki et al. [132] also reported that an increase in phage dosage resulted in the proportional and effective control of colonization by *Salmonella* spp. Using poultry and swine assays in vitro and in vivo, Nale et al. [134] determined the potential of twenty-one myoviruses and one siphovirus in eliminating *Salmonella*. Individual phages significantly reduced the growth of test isolates within six hours post-infection and the subsequent phage administration. However, bacterial regrowth within an hour following treatment suspension was reported, indicative of bacterial resistance to phage therapy. A novel three-constituent phage cocktail was employed in vitro for its lytic efficacy in an optimized *Galleria mellonella* larva model infected with *Salmonella* to remedy the resistance. Comparatively to the individual phages, the cocktail had broader bacterial range coverage, improved lytic efficiency, and prevented the emergence of resistant strains. The study is further supported by Pelyuntha et al. [135] in their comparison of the lytic profile of individual phages and a phage cocktail against *Salmonella* colonization, implicated in the broiler gastrointestinal tract, to enhance poultry consumption safety. Pronounced synergistic enhanced lytic activities and evasion of resistance were obtained with the cocktail compared with the individual agents. The study also affirmed that phages, generally considered safe by the FDA and specific in action, remain potential ideal biocontrol agents for bacteria colonization and biofilm formation in various edible products.

### 6.5. Therapeutic Applications of Bacteriophages against Pathogenic Fusobacterium nucleatum

Pro-tumoral *F. nucleatum* is significant in advancing colorectal cancer and potentially influences the therapeutic response [136]. By incorporating the principles of nanotechnology in a strategic attempt at gut microbiota manipulation, Zheng et al. [136], in demonstration of phage-guided nanotechnology and the potential to control *F. nucleatum* colonization in the gut, drastically improved the treatment of colorectal cancer. The oral or intravenous administration of irinotecan-loaded dextran nanoparticles covalently linked to azide-modified phages inhibited the growth of *F. nucleatum* and thus enhanced the effectiveness of first-line chemotherapy therapies for cancer. Similarly, Dong et al. [137] formulated *F. nucleatum*-binding M13-phage-loaded silver nanoparticles (AgNPs) to achieve targeted clearance of *F. nucleatum* and remodeling of the tumor-immune microenvironment. The in vitro and in vivo studies showed efficient eradication of the bacteria from the gut. Also, significant suppression of the myeloid-derived suppressor cells at the tumor site and the activation of antigen-presenting cells by the M13 phages were observed. These immunomodulatory activities boosted the capacity of the host immune system for colorectal cancer suppression.

### 6.6. Therapeutic Applications of Bacteriophages against Pathogenic Shigella spp.

Shahin et al. [138] determined the efficacy and specificity of individual *Shigella*-specific bacteriophages (vB_SflS-ISF001 and vB_SsoS-ISF002) and a cocktail of both. The phage preparations were investigated against multidrug-resistant *Shigella sonnei* and *Shigella flexneri* isolates. The individual bacteriophages showed high lytic activity in about 75% of the isolates. However, the phage cocktail inhibited 85% of the isolates, indicating higher effectiveness and specificity against a wide range of ESBL-positive and -negative isolates of *S. sonnei* and *S. flexneri*.

### 6.7. Therapeutic Applications of Bacteriophages against Pathogenic Klebsiella pneumoniae

Gut commensals like *K. pneumoniae* opportunistically worsen gut conditions such as inflammatory bowel diseases (IBDs). Federic et al. [139] identified multi-resistant *K. pneumoniae* strains strongly associated with the exacerbation of gastrointestinal disease following enhancement of intestinal inflammation in colitis-prone, germ-free mice challenged with IBD-associated *K. pneumoniae* strains. Stepwise production of lytic cocktails of five-phage targeting strains enabled the effective control of the bacteria in vivo and further supports the use of phage combination therapy in addressing resistance and generally managing gut disease-contributing pathobionts. The lytic effect of some commercially available bacteriophage preparations on strains of *K. pneumoniae* isolated from infants with functional gastrointestinal disorders (FGIDs) was assessed by Grigorova et al. [140] via the drip method and according to clinical recommendations. However, low-level lytic activities and sensitivity to *K. pneumoniae* correlated with age. Significant levels of lysis were observed in children of three to six months but still reflected the inefficiency of this therapy in eliminating *K. pneumoniae* from the intestinal microbiota of children with FGID and suggested that more ingenious and radical approaches to ensuring complete eradication of the associated *K. pneumoniae* are needed.

### 6.8. Therapeutic Applications of Bacteriophages against Pathogenic Listeria monocytogenes

*L. monocytogenes* is a facultative anaerobic Gram-positive bacterium prevalently implicated in foodborne and gastroenteritis diseases [141]. A phage cocktail designated as a foodborne outbreak pill (FOP) and targeted at the implicated *L. monocytogenes* was evaluated by Jakobsen et al. [142] in simulated small intestine, large intestine, and Caco-2 models. Extensive inhibition of *L. monocytogenes* with results comparable to that of a standard drug (ampicillin) was reported. Strikingly, unlike ampicillin, the FOP did not inhibit commensal bacteria in the small intestine, significantly and selectively lysing the *L. monocytogenes* population while being stable in the gastric environment. Furthermore, the FOP prevented the invasion and adhesion of *L. monocytogenes* through a Caco-2 monolayer. Generally, the study highlighted the essential health benefits of phage in this regard and their delivery as dietary supplements, enhancing natural defenses of the gastrointestinal tract against specific foodborne pathogens.

### 6.9. Therapeutic Applications of Bacteriophages against Pathogenic Ruminococcus gnavus

The bacteria *Ruminococcus gnavus*, a Gram-positive anaerobe, is also widely prevalent in the microbiota of humans with inflammatory bowel disease conditions due to Crohn’s disease [143]. Buttimer et al. [144] isolated, characterized, and analyzed six phages that infect the *R. gnavus* JCM 6515T strain. Although no significant decrease in the bacterial count was reported post-phage administration, the study provided insight into two significant mechanisms through which phages interact with *R. gnavus* in the human gut microbiome.

### 6.10. Therapeutic Applications of Bacteriophages against Pathogenic Campylobacter spp.

*Campylobacter*, a major component of the gut microbiota, especially in birds and livestock, is a major foodborne and diarrheal disease-causing bacterial species [145]. D’Angelantonio et al. [146], in their study against *Campylobacter jejuni*, demonstrated colony reduction in a broiler before slaughter, following a two-step phage administration process involving two double-stranded phages (Φ 16-izsam and Φ 7-izsam) belonging to the Caudovirale order. A 0.1 MOI of Φ 16-izsam was administered to a broiler group on the 38th day of rearing, while Φ 7-izsam at an MOI of 1 was administered on the 39th day to another group; these showed a significant one to two log reduction in *C. jejuni* counts on the cecal content compared with the control group after sacrificing the birds on the 40th day. The lowest colony count was, however, observed with an MOI of 0.1 of Φ 16-izsam. Also, Nowaczek et al. [147] isolated 48 strains from 140 broiler chickens (31 *Campylobacter jejuni* and 17 *Campylobacter coli*), which exhibited varying and high-level multi-resistance to the selected antibiotics ciprofloxacin, erythromycin, gentamicin, and tetracycline. They further identified and characterized bacteriophages, including bacteriophages φ4, φ44, φ22, φCj1, φ198, and φ287, placed in the *Myoviridae* and *Siphoviridae* of *Caudovirales* order, and demonstrated the susceptibility of a significant number of the *Campylobacter* spp. to the phage isolates, which had a lytic spectrum of 6, 4, 4, 3, 8, and 7, respectively.

### 6.11. General

Amidst the substantial efforts at identifying effective phages and their cocktails against bacteria-implicated gastrointestinal disorders, it remains desirable to develop effective delivery systems capable of protecting and stabilizing phage particles and products from degradation, destruction, and inactivation under the gastrointestinal tract’s acidic conditions. A smart biocontrol chitosan-encapsulated bacteriophage cocktail formulation was employed by Rbahimzade et al. [148]. The formulation was evaluated as a prophylactic and treatment option for gastrointestinal infections, specifically diarrhea. Experimental animals were challenged with *S. enterica*, *Shigella flexneri*, and *E. coli*, after which treatment commenced with the formulation. Findings reveal that phage encapsulation protected therapeutic life forms from enzymatic degradation, with the non-treated experimental animals experiencing weight loss. However, the highest lytic activity was obtained three days post-phage treatment compared with other studies, where lytic activities took seven to ten days to become evident.

**Table 2 pharmaceutics-15-02416-t002:** Therapeutic applications in the management of certain bacteria-implicated gut diseases.

Target Pathogen	Phage Therapy	Disease	Study Type, Model	Reports	References
*Vibrio* spp.	Cocktail of five (5) lytic Vibrio phages	Fighting *V. cholerae* infection	In vivo, rabbits	Oral administration of cocktails before infection resulted in prophylactic effects. The phage cocktail significantly reduced bacterial load 6 and 12 h after the challenge.	Jaiswal et al. [109]
	Oral phage cocktail therapy	*V. cholerae* infection	In vivo, mice	Phage cocktail (10^8^ PFU/mL) given once daily significantly reduced bacterial load.	Jaiswal et al. [110]
	Bacteriophage pVp-1	Multiple antibiotic-resistant *V. parahaemolyticus* implicated in gastroenteritis	In vivo, mice	Protection from infection and death 1 h after inoculation with *V. parahaemolyticus*.	Jun et al. [111]
	Cocktail of three (3) lytic (virulent) phages—ICP1, ICP2, and ICP3	Cholera pathogenesis/cholera-like diarrhea	In vivo, infant mice and rabbits	Effective at preventing mouse small intestinal colonization. Prophylaxis against the onset of cholera-like diarrhea was achieved after oral administration of the phages up to 24 h before *V. cholera* infestation.	Yen et al. [112]
*Escherichia coli*	Cocktail of three (3) bacteriophages from Siphoviridae and Podoviridae	*E. coli* implicated in gastrointestinal diseases	In vitro	Phage cocktail exhibited broad spectrum and strong lytic activity against *E. coli* isolates.	Nasr-Eldin et al. [113]
	Cocktail of lytic phages specific against *E. coli*	Gut pathogenic *E. coli*	In vivo, mice	Suppression of *E. coli* was observed 5–10 d after phage therapy.	Abdulamir et al. [114]
	Lytic Myoviridae phage phPE42	Extensively drug-resistant (XDR) *E. coli* implicated in foodborne infections	In vivo, rats	Effective eradication of XDR *E. coli* was observed in animal feces.	Abdelaziz et al. [115]
	T4-like phages	Childhood diarrhea-associated *E*. *coli* isolates	In vitro, cultures	T4-like phages combined in a cocktail resulted in increased bacterial lysis.	Bourdin et al. [116]
	Phage-based probiotic dietary supplement consisting of 7 bacteriophage strains	Traveler’s diarrhea (TD) caused by *E. coli, S. flexneri, S. sonnei, S. enterica, L. monocytogenes, S. aureus*	Clinical study, in vivo, humans and mice	Prophylactic effect against TD.	Aleshkin et al. [117]
	Specific bacteriophage	Enteropathogenic *E. coli* (EPEC)	In vivo, mice	A single dose of the phage rendered a protective effect on the bacteria throughout the study.	Vahedi et al. [118]
	T4-like coliphages	Acute bacterial diarrhea	Clinical trial, humans	Failure to improve diarrhea condition, possibly due to insufficient phage concentration.	Sarker et al. [119]
	Virulent bacteriophages targeting prototype of the Adherent Invasive *E. coli* (AIEC) strain LF82	Crohn’s disease (CD)	Ex vivo, in vivo, murine and human intestinal samples	Three virulent bacteriophage cocktails were active against the AIEC strain LF82. A single dose of the cocktail reduced colitis symptoms in mice colonized with AIEC.	Galtier et al. [120]
	Bacteriophage cocktail Ec17B153DK1 vs. the broad-spectrumantibiotic ciprofloxacin	*E. coli* infecting the gut environment	In vitro, simulated small intestine system	The cocktail was effective in reducing *E. coli* in simulated gut conditions. No impact on commensal, non-targeted bacteria.	Cieplak et al. [121]
	Commercial cocktail of *E. coli*-targeting bacteriophages (PreforPro^®^) containing four phages (*LH01-Myoviridae*, *LL5-Siphoviridae*, *T4D-Myoviridae*, and *LL12-Myoviridae*)	Effect on gut microbiota during GI distress and markers of intestinal and systemic inflammation	Clinical trial, humans	The potential of bacteriophages to selectively reduce target organisms without causing dysbiosis	Gindin et al. [122]
	Supplemental bacteriophages (PreforPro^®^)	Enhance the effects of a common probiotic, *B. animalis* subsp. *Lactis* (*B. lactis*) on GI health	Clinical study, humans	Improvements in GI inflammation and colon pain in individuals consuming *B. lactis* with PreforPro^®^.	Grubb et al. [123]
	Suppository containing probiotic strains of *Lactobacillus* spp. and bacteriophages specific for pathogenic *E. coli*	Diarrhea	In vivo, calves	Probiotic-phage suppositories reduced the duration of diarrhea in calves. The complete stopping of diarrhea was observed 24–48 h after use.	Alomari et al. [124]
	Lytic phages (T4, F1, B40-8, and VD13 phages)	Effect on mice gut colonized with human commensal bacteria	In vivo, gnotobiotic mice	Targeted lysis of susceptible gut bacteria. Modulation of non-targeted bacteria through interbacterial interactions.	Hsu et al. [95]
	Genetically engineered temperate phages	Shiga-toxin (Stx)-producing *E. coli* colonizing the mammalian gut	In vivo, mice	Significant repression of fecal Stx concentrations. Suppression of virulence factors in gut bacteria.	Hsu et al. [125]
	Phage PDX, a member of the *Myoviridae* family	Diarrheagenic enteroaggregative *E. coli* (EAEC)	In vitro, in vivo, cultures and mice	Bacteriolytic activity of EAEC isolates (EN1E-0007) in vitro and in vivo. No dysbiosis was observed in the anaerobic culture.	Cepko et al. [126]
	Phage ES17, a Podoviridae phage	Extraintestinal pathogenic *E. coli* (ExPEC) in the intestine	In vivo, mice	Selective elimination of invasive pathobiont species from mucosal surfaces in the intestinal tract.	Green et al. [128]
*Clostridium difficile*	Six (6) myoviruses and one (1) siphovirus	*C. difficile* infection (CDI)	In vitro, in vivo, hamsters	Specific phage combinations resulted in total lysis of *C. difficile* in vitro. Prevention of resistance. In vivo, the evaluation revealed a reduction in *C. difficile* colonization 36 h post-infection.	Nale et al. [129]
	Recombinant bacteriophage	*C. difficile* infection	In vitro, in vivo, cultures and mice	Targeting and killing of *C. difficile*.	Selle et al. [130]
*Salmonella* spp.	Phage SE20 (Podoviridae)	*S. enterica* serotype Enteritidis	In vitro, in vivo, mice	Oral administration of a single dose of bacteriophage protected against salmonellosis and treatment of salmonellosis. Animals developed hepatomegaly and splenomegaly as side effects but had no gastrointestinal complications with the phage therapy.	Dallal et al. [131]
	Bacteriophage cocktail (foodborne outbreak pill (FOP) targeting *E. coli* O157:H7, *L. monocytogenes*, and *Salmonella*)	*Salmonella* infection	In vitro	Simulator of the Human Intestinal Microbial Ecosystem (SHIME).	Moye et al., 2019 [133]
	Phage cocktail	*Salmonella* colonization in experimentally challenged birds	In vivo, birds	Phage treatment effectively reduced Salmonella colonization and enhanced growth performance weight gains in challenged birds.	Thanki et al. [132]
	Myoviruses and a siphovirus	*Salmonella* infection gastrointestinal enteritis	In vitro, in vivo, swine, birds, cultures	Phage cocktail (STW-77 and SEW-109) had the most lysing efficacy on the swine and bird models. Some phages from the cocktail could lyse resistant strains of the organism.	Nale et al. [134]
	*Salmonella* phages (vB_SenS_KP001, vB_SenS_KP005, and vB_SenS_WP110)	*Salmonella* colonization in the gastrointestinal tract of broilers	In vivo, broilers	The phage cocktail reduced Salmonella colonization in broilers’ gastrointestinal tracts from over 70% to 0% 4 d post-treatment.	Pelyuntha et al. [135]
*Fusobacterium nucleatum*	Irinotecan-loaded dextran nanoparticles covalently linked to azide-modified phages.	Colorectal cancer (CRC)	In vivo, mice	Phage administration inhibited the growth of *F. nucleatum*. It significantly boosted the effectiveness of first-line chemotherapy treatments for CRC.	Zheng et al. [136]
	*F. nucleatum* (*Fn*)-binding M13-phage-loaded silver nanoparticles (AgNPs)	Symbiotic *F. nucleatum* in the gut selectively increases immunosuppressive myeloid-derived suppressor cells (MDSCs), thereby promoting colorectal cancer (CRC) progression.	In vitro, in vivo, mice	Treatment with M13-phage-loaded AgNPs could mop up *F. nucleatum* in the gut, resulting in non-amplification in MDSCs at the tumor sites.	Dong et al. [137]
*Shigella* spp.	*Shigella*-specific bacteriophages: vB_SflS-ISF001, vB_SsoS-ISF002, and a cocktail of both	*S. sonnei* and *S. flexneri* causing human acute gastrointestinal infections	In vitro, cultures	More than 85% of the ESBL-positive and -negative isolates of *S. sonnei* and *S. flexneri* were inhibited by the phage cocktail (vB_SflS-ISF001 and vB_SsoS-ISF002.)	Shahin et al. [138]
*Klebsiella pneumoniae*	Lytic five-phage combination	Inflammatory bowel disease (IBD)-associated *K. pneumoniae* (Kp) strains	In vivo, mice	Suppression of colitis in mice	Federic et al. [139]
	Commercial bacteriophage preparations	*K. pneumoniae* strains isolated from children with functional gastrointestinal disorders (FGIDs)	In vitro, spot test	Phages show negligible lytic activity, indicating the need for a more radical approach to eradicating *K. pneumoniae* in children with FGIDs.	Grigorova et al. [140]
*Listeria monocytogenes*	Bacteriophage cocktail (Foodborne Outbreak Pill (FOP))	*L. monocytogenes*	In vitro, simulated ilium and colon conditions	Protection against *L. monocytogenes* infecting the human gastrointestinal tract without causing dysbiosis.	Jakobsen et al. [142]
*Ruminococcus gnavus*	Six bacteriophages	Mucin-degrading bacterium *R. gnavus* from the human gut	In vivo, mice	Results show the coexistence of phages with *R. gnavus* in the human gut microbiome.	Buttimer et al. [144]
*Campylobacter* spp.	Double-stranded phages (Φ 16-izsam and Φ 7-izsam)	*C. jejuni* associated with broilers	In vivo, broilers	Phage administration showed a significant one to two log reduction in *C. jejuni* counts on the cecal content compared with the control group after sacrifice. The lowest colony count was, however, observed with an MOI of 0.1 of Φ 16-izsam.	D’Angelantonio et al. [146]
	Bacteriophages φ4, φ44, φ22, φCj1, φ198, and φ287	*C. jejuni* associated with broilers	In vitro, in vivo, broilers	Demonstrated the susceptibility of a significant number of the multi-resistant *Campylobacter* spp. to the phage isolates, which had a lytic spectrum of 6, 4, 4, 3, 8, and 7, respectively.	Nowaczek et al. [147]
General	Chitosan-encapsulated bacteriophage cocktail	*S. enterica*, *S. flexneri*, and *E. coli* gastrointestinal infections	In vivo, rats	Reduction in positive cultures from stools of the group receiving the chitosan-encapsulated bacteriophage cocktail was observed after two days.	Rahimzade et al. [148]

## 7. Roles in Preventing and Treating Specific Gut Microbiota-Related Diseases

Many benefits are associated with the use of bacteriophages in the treatment of bacterial-related diseases in comparison with conventional antibiotics. Apart from being safe, bacteriophage actions are specific and, thus, retain the homeostatic nature of the gut microbiota following their use. Also, since they hijack the host’s replicative machinery within the host, there is usually no need for successive administrations, as they utilize the host for their propagation until their exhaustion, at which point the therapeutic aim is achieved [149,150]. They play different roles in preventing and treating specific gut microbiota-related diseases via the different mechanisms described above and summarized in Figure 3. In controlling gastrointestinal diseases, phages can infect, reduce bacterial loads, and eliminate implicated pathogens, including antibiotic-resistant ones. They also aid in restoring the gut microbiota, alleviating symptoms, and promoting healing. The gastrointestinal tract’s inflammatory burden, including ulcerative and Crohn’s disease, can be managed through microbiota modulation, biofilm disruption, and the control of proinflammatory bacterial flare-ups [120]. Dysbiosis interrupts the bacterial population with a consequential impact on metabolic processes, including glucose and fatty acid metabolism, adiposity leading to obesity, non-alcoholic fatty liver, and heart diseases [151]. Bacteriophage-aided gastrointestinal tract microbiota optimization is instrumental for efficient metabolic conditions. The bacteriophages’ ability to modulate both the innate and adaptive immune system is a veritable tool in the fight against cancer [152] and opens up a new frontier for their possible application in diagnosing and treating cancer [153]. Also, certain bacteria species have been significantly linked to the development of some forms of gastrointestinal cancers. Typically, colorectal and pancreatic cancers have significantly been linked to certain bacteria, including *F*. *nucleatum* and *Porphyromonas* spp., for which studies have reported bacteriophages with anti-colorectal and -pancreatic cancer potentials, with promising results [154,155,156]. Finally, bacteriophages contribute to the function of the gut–brain axis communication as it relates to gut microbiota regulation, immunomodulation, and metabolite production and have applicable uses against neurological disorders [157]. For example, Ghadge et al. [158] demonstrated a reduction in motor neuron loss, microgliosis, superoxide dismutase 1 (SOD1) burden and aggregation, and astrocytosis through the use of phage-specific single-chain variable fragment antibodies (scFvs) raised against SOD1^G93A^ in mice.

## 8. Conclusions

Lytic bacteriophages are abundant inhabitants within the gut microbiota, and they are engaged in constant interactions with gut-associated bacteria, which act as their hosts, ensuring their propagation. These interactions have far-reaching implications for human health, as phages play essential roles in the gut system and overall well-being. Extensive research has implicated them in managing various gut-related diseases, though most of these findings are still in the early stages of development. Translating these research outputs into clinical applications poses a complex challenge, necessitating the synergistic engagement of related stakeholders in initiating and evaluating clinical trials, advancing research, and investigating potential clinical applications and commercialization. The significance of understanding the roles of bacteriophages in the gut and human health cannot be overstated.

One crucial aspect is their role in effectively regulating gut microbiota composition. Phages target specific bacterial species, either promoting the growth of beneficial bacteria or controlling the overgrowth of harmful ones. This delicate balance is vital for maintaining a healthy gut microbiota, which is essential for various physiological functions and overall health. Nutrient digestion and absorption are intricately linked to the gut microbiota’s activities, and bacteriophages play a significant role in this process. The gut microbiota breaks down complex carbohydrates, synthesizes specific vitamins, and aids nutrient absorption. Phages’ influence on the gut bacterial community affects these processes, which can have implications for nutrient availability and utilization by the host. Imbalances in the gut phage–bacterial community could lead to inefficient digestion and nutrient absorption, potentially contributing to malnutrition or other health issues. Another critical aspect of gut phage–bacterial interactions is their impact on the modulation of the immune system. The gut is a critical interface between the external environment and the body’s internal systems. The gut microbiota, including bacteriophages, is pivotal in educating and shaping the immune system and can influence the bacterial communities in the gut, leading to changes in immune responses.

A well-balanced gut phage–bacterial community can help prevent inappropriate immune reactions, reducing the risk of autoimmune disorders and allergies. They also hold the potential as a selective tool to combat gut bacterial pathogens, particularly multi-resistant ones. The rise of antibiotic-resistant bacteria has become a significant health concern worldwide, necessitating the exploration of alternative treatment options. Phage therapy, which involves using specific phages to target and kill pathogenic bacteria, offers a promising approach to addressing this challenge. Understanding the intricate interactions between bacteriophages and gut bacteria is crucial for developing personalized and targeted phage therapies that are both safe and effective.

Furthermore, dysbiosis, an imbalance in the gut phage–bacterial community, has been associated with various diseases, including inflammatory bowel diseases (IBDs), gut-related cancers, and metabolic disorders. Research on phages and their role in managing dysbiosis-related diseases holds immense potential for providing insights into disease development and potential therapeutic strategies. The intrinsic interactions between bacteriophages and host bacteria are a foundation for advancing personalized and targeted phage therapies. This emerging field of research aims to utilize phages as precision medicine tools, where specific phages are selected based on individual gut microbiota profiles and health conditions. Such personalized therapies have the potential to revolutionize disease treatment, particularly in cases where traditional treatments have shown limited efficacy.

In conclusion, understanding the roles of bacteriophages in the gut and human health is of utmost importance. These tiny viral agents play pivotal roles in regulating gut microbiota composition, nutrient digestion, immune system modulation, and protection against pathogens. Their potential in managing gut-related diseases, including dysbiosis, offers promising avenues for future therapeutic interventions. Collaboration among various stakeholders is essential to translating research findings into clinical applications. By harnessing the power of bacteriophages, we can unlock new and innovative approaches to improve human health and combat challenging diseases associated with the gut microbiota.

## Figures and Tables

**Figure 1 pharmaceutics-15-02416-f001:**
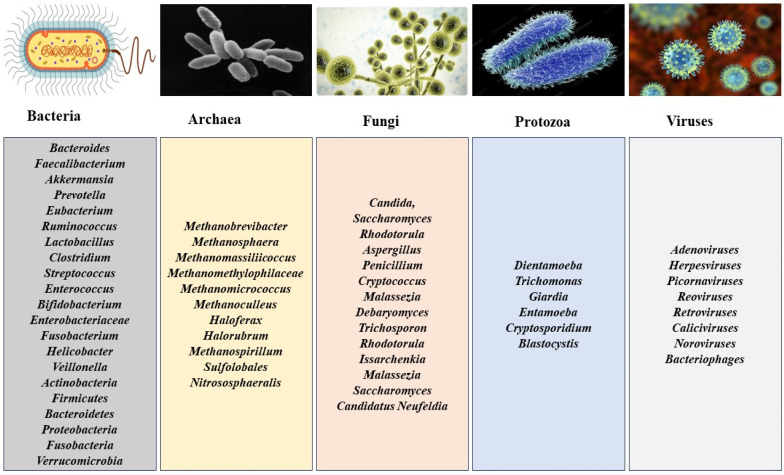
The microbial genera/order composition of the gut microbiota.

**Figure 2 pharmaceutics-15-02416-f002:**
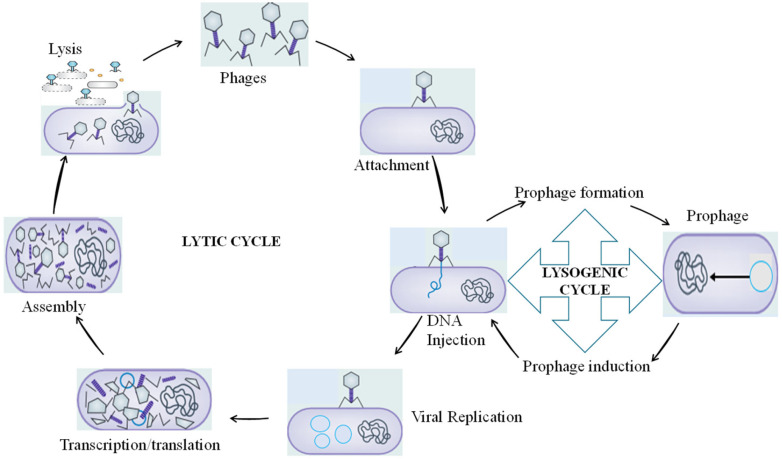
The lytic or the lysogenic pathways of a phage lifecycle.

**Figure 3 pharmaceutics-15-02416-f003:**
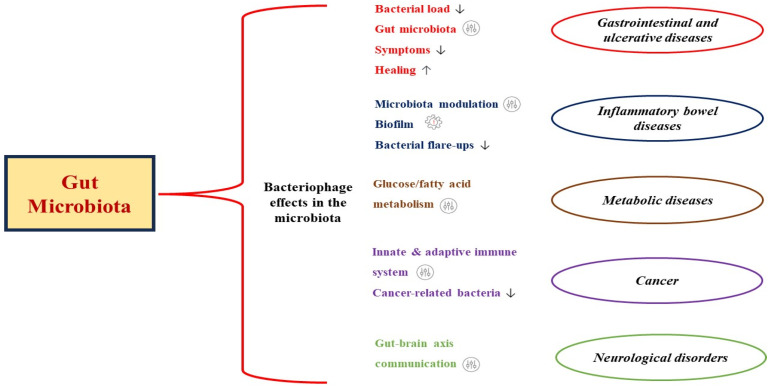
Roles of bacteriophage interactions in the gut in preventing and treating specific gut microbiota-related diseases.

**Table 1 pharmaceutics-15-02416-t001:** Bacteriophage classifications.

Morphology	Genomic Properties	Life Cycle
Caudovirales ∘ Myoviridae ∘ Siphoviridae	DNA Phages ∘ Double-stranded DNA (dsDNA) Phages ∘ Single-stranded DNA (ssDNA) Phages	Temperate
Filamentous	RNA Phages ∘ Double-stranded RNA (dsRNA) Phages ∘ Single-stranded RNA (ssRNA) Phages	Lytic
Tectiviridae	Retroviruses	Lysogenic
Inoviridae	Circular Replicating Phages	
Leviviridae	Temperate Phages	
Microviridae	Virulent Phages	
Pleolipoviridae		

## Data Availability

Data sharing not applicable.

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
