# Peer review of "The Role of Bacteriophages in the Gut Microbiota: Implications for Human Health"

_pharmaceutics, 2023, doi:10.3390/pharmaceutics15102416_

Round 1

Reviewer 1 Report

Many small English errors which cause some difficulties. If a revised version is acceptable for publication I would be happy to correct the English.

Author Response

Comments and Suggestions for Authors

  • I found this review interesting and covers a topic of great importance. I was rather hoping that it would be a detailed exploration of the interaction of “commensal” phage with commensal components of the gut flora but, as the authors mentioned, there is not so much known about this so, quite understandably, they include the effect of phage on pathogens.

The bacteriophage interactions were done through the application of bacteriophages against the disease-causing bacteria. These where used in precipitating the outcome of such interaction as it relates to the resulting disease conditions. This is justifiable, as bacteriophage effect can not be discussed without the focus on their lytic effects against bacteria.

  • As a review I do not think is acceptable for publication as it is currently and I would like the authors to consider making the following major changes.

Thank you for the kind suggestions.

  • Some of the sections were too long. The introduction could be reduced by 50% since much of what is said is repeated in the following sections. The section on bacteriophages is also unnecessarily long. Again, although this is just my opinion, it could be reduced by 50 % since this review should describe the interactions rather than be a study of phage itself.

These sections have been adjusted, significantly reducing the text.

  • I was disappointed in the selection of papers in the biography. Most of the papers were recent, which is fine, but there is little mention of earlier work on which the more recent work is based. For example, there is no mention of the seminal work by Williams Smith in 1980s using phage in pigs, lambs and calves. One paper is mentioned about treatment of calves. I know this paper and the bibliography is also highly selective – well this is a problem of inadequate knowledge of the literature. For example, why mention Thank et al 2023 (164) and not Atterbury et al. (2007) Appl. Env. Micro. 73.4543? I feel that the authors should trawl through the various parts of the literature and also find earlier more seminal references on which many of the papers mentioned are based. In some of this early work (particularly Smith) there is a clear rational for phage choice to avoid the too great specificity seen with LPS-specific phages i.e. using phage targeting surface virulence determinants such that any phage-resistant mutants that arise will be less virulent.

Dear, your consideration is very valid and well respected, however, the data sourcing considered only the published materials in the last ten years (2013 - 2023). This statement has also been included in the methodology used on the review.

  • Page 1-2. The sentence is 12 lines long. It could be broken up using (i), (ii) … etc for the various sections to make it more readable.

The unnecessary and lines, including those stating common knowledge has been removed to shorten the section.

  • 3. This is too specific. Many phages do not have this morphology. I suggest you delete the figure.

The figure has been deleted as recommended.

  • Section 6 p.10 lines 6-11 are repetition.

The repeated lines have been removed as recommended. Thanks.

  • Reference 157. There is also fairly large literature on various methods to control or eliminate plasmid-mediated AMR including the use of plasmid-specific phages. It may be that the authors do not want to include this area of work but it is relevant to the gut and some of the in vivo work shows high efficacy.

The review method involved an overview of recent literature using the relevant databses, including PubMed, Scopus, Google Scholar, and Thompson Reuters, using the keywords including ‘Bacteriophages’, ‘Gut Microbiota’, ‘Human Health’ and applying Boolean connectors were needed. Subscription-based articles were however, not considered in the review. Thus, we may have missed these following the exclusion criteria.

  • A lot of the references related to in vivo work are using mouse model, For some pathogens this is all that is available but not for the major pathogens E. coli and Salmonella. How about Campylobacter where some good work has been done.

In agreement, most of the literature in vivo mouse studies, however, others, including humans, rabbit, swine, birds, calves, Simulated ilium and colon condition, in vitro, and culture-based studies were reported.

Relevant articles reporting Campylobacter spp has been added to the manuscript.

  • The authors do recognise the difficulties of treating gastro-intestinal infections with phage as there may be several causes of infection and more than one present in the infection. It is an important point.

Thanks.

  • Is Table 2. Needed given that much of this information is in the text. It should be one or the other. The Table is very clear and I would favour reducing the text length and referring to the table.

The studies in the literature involving therapeutic application of phages in details were reported in the section and summarised in Table 2.

Reviewer 2 Report

Authors evaluated the “The Role of Bacteriophages in the Gut Microbiota: Implications for Human Health”.

1- The text of the study is very long and it is useful to focus on "bacteriophages", which is the main purpose of the study.

2- The study is about bacteriophages and their therapeutic efficacy however the first 7 pages of the article include general microbiota composition and related factors. The section on factors influencing the microbiota is very limited and does not naturally include many factors (can be omitted unless factors influencing the composition of the gut bacteriophage are discussed). It would be useful to summarise these sections and remove most of them. Similarly, Figures 1 and 2 are related to the general microbiota composition and their inclusion in this bacteriophage-specific study does not seem necessary. Figure 1 is too simple and limited to be included in an scientific article on microbiota or bacteriophages (it should be kept in mind that the microbiota composition outside these anatomical regions is previously described in detail). In this way, it will be possible to read and follow the content of the article more easily. Considering that the researchers used nearly 100 references in this section, summarising this section will also reduce the number of references (to reasonable levels).

3- The section on bacteriophage identification and classification is very well written and it would be much better to start the article with this section.

4- Although the researchers described the microbiota composition in detail in the early parts of the study, the intestinal virome (and hence bacteriophage content) and related factors were not discussed in the study. Similarly, breast milk is the most influential factor on the microbiota in infants in the first 1000 days of life, and breast milk virome content (especially bacteriophages) was not discussed in this study. If the investigators are going to evaluate only phage therapy approaches, it would be more appropriate to exclude all general microbiota information. But in any case, it would be more appropriate to examine the human virome/bacteriophage composition and the factors affecting it.

5- In the study, experimental, preclinical or clinical studies on the efficacy of phage therapies were given, but data on safety were not shared. It would be useful for researchers to share data on safety, adverse effects and concernss. 

6- Finally, Figure 5 should be reviewed, and  the relationship between bacteriophages and microbiota is mentioned on the table should be reviewed in the text or in the figure with references.

In conclusion, although the researchers have prepared a very detailed text on a current and important issue, it would be useful for them to reconstruct the text in line with the message they want to convey (as stated in the title)

Author Response

Author Comments and Suggestions

Authors evaluated the “The Role of Bacteriophages in the Gut Microbiota: Implications for Human Health”.

1- The text of the study is very long and it is useful to focus on "bacteriophages", which is the main purpose of the study.

The text has been adjusted accordingly as suggested.

2- The study is about bacteriophages and their therapeutic efficacy however the first 7 pages of the article include general microbiota composition and related factors. The section on factors influencing the microbiota is very limited and does not naturally include many factors (can be omitted unless factors influencing the composition of the gut bacteriophage are discussed). It would be useful to summarise these sections and remove most of them. Similarly, Figures 1 and 2 are related to the general microbiota composition and their inclusion in this bacteriophage-specific study does not seem necessary. Figure 1 is too simple and limited to be included in an scientific article on microbiota or bacteriophages (it should be kept in mind that the microbiota composition outside these anatomical regions is previously described in detail). In this way, it will be possible to read and follow the content of the article more easily. Considering that the researchers used nearly 100 references in this section, summarising this section will also reduce the number of references (to reasonable levels).

The section on factors influencing the microbiota has been omitted as recommended.

Figure 1 has been removed as recommended. The section 2 has been summarised.

3- The section on bacteriophage identification and classification is very well written and it would be much better to start the article with this section.

Thanks

4- Although the researchers described the microbiota composition in detail in the early parts of the study, the intestinal virome (and hence bacteriophage content) and related factors were not discussed in the study. Similarly, breast milk is the most influential factor on the microbiota in infants in the first 1000 days of life, and breast milk virome content (especially bacteriophages) was not discussed in this study. If the investigators are going to evaluate only phage therapy approaches, it would be more appropriate to exclude all general microbiota information. But in any case, it would be more appropriate to examine the human virome/bacteriophage composition and the factors affecting it.

In the ‘Bacteriophages: Classification, Life Cycle, and General Mechanism of Action’ section, the different bacteriophage classifications were done, and being ubiquitous are present in the gut. we also highlighted Caudovirales are the most common and prominent class.

Some of the reviewers suggested we focus on the impact of bacteriophages in human health and remove the factors. The factors were however highlighted in the introductory part including the suggested addition of the impact of breastmilk on infants: ‘… breast milk have significant influence on the infant microbiota in the first few years…’

5- In the study, experimental, preclinical or clinical studies on the efficacy of phage therapies were given, but data on safety were not shared. It would be useful for researchers to share data on safety, adverse effects, and concerns.

The review focus was chiefly on the impact of bacteriophages in the gut microbiota and the implications for human health. This we did through the effects against the relevant bacteria. Notwithstanding, most safety, adverse effects, and concerns of using phages are common knowledge., through review of current data around it, could easily make a new manuscript.

6- Finally, Figure 5 should be reviewed, and the relationship between bacteriophages and microbiota is mentioned on the table should be reviewed in the text or in the figure with references.

The figure 3 (formally figure 5), represent the roles of bacteriophage interactions in the gut in preventing and treating specific gut microbiota-related diseases. The symbols representing relationships, including the upregulation, down regulation, modulation, and process disruption as it relates to the specific gut microbiota-related diseases.

In conclusion, although the researchers have prepared a very detailed text on a current and important issue, it would be useful for them to reconstruct the text in line with the message they want to convey (as stated in the title)

Text has been adjusted as suggested by the reviewers.

Reviewer 3 Report

This manuscript describes an interesting review on the role of bacteriophages in the gut microbiota with implications for health of humans.

General remark: A manuscript published in a journal in the first quarter according to the journal impact factor should be careful to write the correct and up-to-date terms with regard to microbiology. Also, a more in-depth and well-structured manuscript is necessary.

In the attacment are some examples of remarks to improve the manuscript:

Author Response

Comments and Suggestions for Authors

  • What is the methodology of literature review? It should be clearly stated how the review was conducted even if it is not a systematic review. Is this a scoping review, narrative review, or perhaps an overview of recent literature?

‘The review method involved an overview of recent literature using the relevant databses, including PubMed, Scopus, Google Scholar, and Thompson Reuters, using the keywords including ‘Bacteriophages’, ‘Gut Microbiota’, ‘Human Health’ and applying Boolean connectors were needed’.

This has been placed in the manuscript.

  • Page 2,3,12,13: the term ‘intestinal flora’ should be replaced with the correct term ‘microbiota’.

These have been adjusted accordingly as recommended.

  • Novel nomenclature should be used for phyla. See: 10.1099/ijsem.0.005056. Firmicutes are now Bacillota, Proteobacteria are now Pseudomonadota, Actinobacteria are now Actinomycetota, Bacteroidetes are now Bacteroidota etc. these new names should be mentioned in the text.

The current suggested names have been duly placed.

  • If not using latin name for bacteria, then italics should not be used as on page 2 of the manuscript for bififobacteria.

This has been adjusted.                                                  

  • Up-to-date terminology should be used for lactobacilli (Zheng et al., 2020, ‘A Taxonomic Note on the Genuc Lactobacillus: Description of 23 Novel Genera, Emended Description of the Genus Lactobacillus Beijerinck 1901, and Union of Lactobacillaceae and Leuconostocaceae’ International Journal of Systematic and Evolutionary Microbiology 70(4):2782-2858. The genus Lactobacillus has been been divided.

Changes has been made were applicable.

  • Clostridium difficile has also been renamed to Clostridiodes difficile.

The suggested new name has been placed.

  • The term ‘probiont’ is uncommon in human use. In Pubmed 62 articles, related only to animal use of probitoics or probiotic candidates use this term. The correct reference for the definition of probiotics as “live microorganisms, that, when administered in adequate amounts, confer a health benefit on the host” (Hill, et al. 2014 https://www.nature.com/articles/nrgastro.2014.66). This is a consensus statement published by the International Scientific Association for Probiotics and Prebiotics (ISAPP) in the journal Nature Reviews Gastroenterology and is based n the original definition that was published in 2001 by the WHO and FAO.

The ’probiont’ term has been corrected in the manuscript.

  • Figures 1 and 2 are quite amicable. This information is not new and is well-known.

Figure 1 has been expunged. Figure 2 is aggregated the gut compositions as collated from different sources, the detailed description in text as been removed and only directed in the figure.

  • Chapter 7 focuses on potential therapeutic applications of bacteriophages. However, caution should be noted. First of all, the subchapters should not be just the names of te potential pathogens as this is misleading. The headings could be for example: “Therapeutic applications of bacteriophages against pathogenic Escherichia coli strains. This is important as our intestinal microbiota also contains several commensal and even beneficial strains of Escherichia coli. One strain of human origin is even a probiotic: Escherichia coli Nissle 1917. Also, the first paragraph 7.1 is not associated with Vibrio spp. Therefore, a much more comprehensive and exact explanation should be added regarding these applications.

The subchapters of the chapter 6 (formerly chapter 7) have all been corrected. Using ‘Therapeutic applications of bacteriophages against pathogenic’ before the potential pathogen.

The first paragraph of 6.1 (former 7.1) has adjusted to focus on Vibrio spp

Reviewer 4 Report

Known in the field based on previous literatures:

1. The collection of microorganisms (bacteria, archaea, fungi and viruses) colonising the gastrointestinal (GI) tract is termed the 'gut microbiota'. The gut microbiota play important role, exists symbiotically, and helps support energy harvesting, digestion, and immune defense.

2. Gut microbiota is influenced by antibiotics, genetic profile, pollutants, and age. Diet is considered as one of the main drivers in shaping the gut microbiota across the life time. Altered gut bacterial composition (dysbiosis) has been associated with the pathogenesis of many inflammatory diseases and infections. 

In this review author’s discussed following findings:

I have gone through the review article titled ‘The Role of Bacteriophages in the Gut Microbiota: Implications for Human Health’. The present article aims to explore the interactions of bacteriophages with bacterial communities in the gut and their current and potential impacts on human health. Authors mentioned the following findings-

1. Authors nicely enlighten about gut microbiota composition and their functions. They also explained the factors which affect gut microbiota composition.

2. Further, they also educated about potential applications of phages for preventive and curative purposes. Various studies revealed that developing therapies against the most prevalent bacteria associated with gut system infections, including Vibrio, Escherichia coli, Clostridium, and Salmonella species.

The article presented is interesting and generally supportive of the conclusions drawn. However, there are some issues that require the authors' attention. The following minor suggestions if incorporated could help in the better understanding of the significance of the study and implications.

 Minor Concerns:

 1. Several articles are available online. How your review article is different from rest? Does it address a specific gap in the field?

2. Authors should also discuss about specific improvements regarding the methodology?

3. The roles of microbiota in neurological disorders are also documented. For better understanding, authors should also discuss the role of microbiota in neurological disorders.

Author Response

Minor Concerns:

  1. Several articles are available online. How your review article is different from rest? Does it address a specific gap in the field?

The manuscript among the above, aggregated the recent research advances against the listed microorganisms and presents basis for further advancement

  1. Authors should also discuss about specific improvements regarding the methodology?

The metholdoyg as heeb added. The stat,ent, ‘The review method involved an overview of recent literature using the relevant databses, including PubMed, Scopus, Google Scholar, and Thompson Reuters, using the keywords including ‘Bacteriophages’, ‘Gut Microbiota’, ‘Human Health’ and applying Boolean connectors were needed’, has been added to the manuscript..

  1. The roles of microbiota in neurological disorders are also documented. For better understanding, authors should also discuss the role of microbiota in neurological disorders.

The role in neurological disorders have been discussed. The following statement was added in the section:

Finally, bacteriophages contribute to the function of the gut-brain axis communication as it relates to gut microbiota regulation, immunomodulation, and metabolite production and have applicable uses against neurological disorders (166). For example, Ghadge et al. (167) demonstrated the reduction of motor neuron loss, microgliosis, superoxide dismutase 1 (SOD1) burden and aggregation and astrocytosis thought the use of phage-specific raised single-chain variable fragment antibodies (scFvs) against SOD1G93A in mice.

Round 2

Reviewer 2 Report

Although the authors have made corrections to the text, there are lot of information which contains general microbiota descriptions and is not relevant to the objective of this article. The authors have chosen to retain this section and the 100 references cited in this section. However, recent format is better than previous one and include major findings related bacteriophages.

Reviewer 3 Report

The manuscript is improved. One note regarding the correction of Clostridium difficile. As noted in review report round 1 the new name is Clostridioides difficile, not Clostridiodes difficile as has been corrected in the manuscript. Please correct throughout the whole manuscript.